# Quantitative Shape-Classification of Misfitting Precipitates during Cubic to Tetragonal Transformations: Phase-Field Simulations and Experiments

**DOI:** 10.3390/ma14061373

**Published:** 2021-03-12

**Authors:** Yueh-Yu Lin, Felix Schleifer, Markus Holzinger, Na Ta, Birgit Skrotzki, Reza Darvishi Kamachali, Uwe Glatzel, Michael Fleck

**Affiliations:** 1Metals and Alloys, University of Bayreuth, Prof.-Rüdiger-Bormann-Straße 1, 95447 Bayreuth, Germany; yueh-yu.lin@uni-bayreuth.de (Y.-Y.L.); felix.schleifer@uni-bayreuth.de (F.S.); markus.holzinger@uni-bayreuth.de (M.H.); uwe.glatzel@uni-bayreuth.de (U.G.); 2Max-Planck-Institut für Eisenforschung GmbH, Max-Planck-Straße 1, 40237 Düsseldorf, Germany; tagetacpu@163.com (N.T.); reza.kamachali@bam.de (R.D.K.); 3School of Material Science and Engineering, University of Science & Technology Beijing, Beijing 100083, China; 4Federal Institute for Materials Research and Testing (BAM), 12205 Berlin, Germany; birgit.skrotzki@bam.de

**Keywords:** phase-field simulation, misfitting precipitate, nickel-base alloy, γ″ phase, Al-Cu alloy, θ′ phase, precipitate shape

## Abstract

The effectiveness of the mechanism of precipitation strengthening in metallic alloys depends on the shapes of the precipitates. Two different material systems are considered: tetragonal γ′′ precipitates in Ni-based alloys and tetragonal θ′ precipitates in Al-Cu-alloys. The shape formation and evolution of the tetragonally misfitting precipitates was investigated by means of experiments and phase-field simulations. We employed the method of invariant moments for the consistent shape quantification of precipitates obtained from the simulation as well as those obtained from the experiment. Two well-defined shape-quantities are proposed: (i) a generalized measure for the particles aspect ratio and (ii) the normalized λ_2_, as a measure for shape deviations from an ideal ellipse of the given aspect ratio. Considering the size dependence of the aspect ratio of γ′′ precipitates, we find good agreement between the simulation results and the experiment. Further, the precipitates’ in-plane shape is defined as the central 2D cut through the 3D particle in a plane normal to the tetragonal c-axes of the precipitate. The experimentally observed in-plane shapes of γ′′-precipitates can be quantitatively reproduced by the phase-field model.

## 1. Introduction

Precipitation strengthening is an important mechanism for the development of new metallic materials with outstanding thermo-mechanical properties [1,2]. The effectiveness of the strengthening mechanism depends on details of the precipitation microstructure. Nowadays, highly specialized aging heat treatments target very specific microstructural features providing materials with optimal mechanical properties [3].

An important quantity is the misfit, which relates to the crystallographic lattice difference between the precipitate and the surrounding matrix phase. Misfitting precipitates cause a certain stress field in their direct environment, which depends on the shape of the precipitate as well as positions of neighboring precipitates. Macroscopic plastic material degradations are expressed by microscopic migrations of dislocations driven by local stresses. Thus, via their stress imprints, the precipitate particles can act as obstacles for mobile dislocations in the matrix. Depending on various characteristics, such as the precipitate volume fraction, the sizes and shapes of the particles as well as the overall distribution of the precipitates, a particular precipitation microstructure can be very effective in trapping dislocations, leading to a significant strengthening effect of the overall material [4].

In this work, we consider two different technologically successful materials with certain common aspects in their precipitation microstructure: (i) the precipitation of the tetragonal γ″ phase (Ni_3_Nb) in Nb-containing Ni-based alloys [5,6] and (ii) the precipitation of the tetragonal θ′ phase (Al_2_Cu) in Al-Cu alloys [7,8]. Both cases involve cubic to tetragonal transformations with a strongly anisotropic lattice misfit, where the misfit, ε_3_, in the tetragonal c-axes is about an order of magnitude larger than the misfit, ε_1_, in the other two basic lattice directions. Furthermore, in both cases, the tetragonal c-axis of the precipitate phase is always found to be strictly aligned with one of the three different crystallographic directions of the cubic matrix phase. The anisotropic lattice misfit results in strongly anisotropic precipitate shapes, as the anisotropic misfit stresses are smaller for a precipitate–matrix interface with a normal pointing in the direction of the tetragonal axis. Therefore, plate-shaped precipitates are obtained, where the plate-normal points in the direction of the tetragonal c-axis.

Figure 1 shows a schematic representation of three different elastically interacting tetragonal precipitates, which are coherently embedded in a cubic matrix phase. The precipitate-matrix interface is indicated by the thick black lines. As thin black lines, we also visualize the elastic deformations, which surround the particles and result from their coherent embedding and the anisotropic misfit. Within this article, the size of a plate-shaped precipitate is quantified by the major diameter *d*, as shown in Figure 1. Further, we define the thickness *w* of the particles, which also relates to the particles’ aspect ratio.

Further, we mention that the γ″ precipitates as well as the θ′ precipitates exist in three different orientational variants [6,8]. Within typical precipitation microstructures, all the three variants can be found at the same time. Thus, elastic particle–particle interactions lead to very complex stress stats in these microstructures. In Figure 2, we exemplarily show the γ/γ′′ microstructure in a Nb-containing Ni-based alloy. Numerous individual γ′′ particles are visible with all three orientation variants.

In both material systems, the precipitates are in metastable phases, which eventually transform into stable but kinetically unfavorable phases, such as the δ phase (Ni_3_Nb) or θ phase (Al_2_Cu). The two metastable precipitate phases can both be obtained by time- and temperature-controlled aging treatment with a rapid cooling process before transforming into final stable phases. These misfitting particles are frequently observed to nucleate heterogeneously on the defects, especially dislocations, due to associated lower nucleation barrier [9,10]. However, even though these two phases share many similarities, the differences in the interfacial energy and lattice strain result in distinguishable microstructures, which will be investigated in this work.

The phase-field method, based on a thermodynamic function of integral over a local phenomenological potential energy density, is a useful tool for modeling solid-state phase transformations [11,12]. This phenomenological energy density may be taken into account by phase-boundary energy, multi-component solute distribution, and elastic strain caused by external loads or lattice misfit between the phases. By general variational principles, a consistent set of coupled partial differential equations that describes the kinetics of the microstructure evolution in the chosen configuration can be derived. The phase-field method is frequently used to study the shape evolution of γ/γ′ microstructures [13,14] as well as systems with γ/γ′/γ′′ microstructure [15]. The variational formulation of the phase-field method makes it a useful tool to determine precipitate shapes of minimum energy [16,17,18].

A general way for the quantification of the shapes of precipitates is given by the method of moment invariants [19]. The method has been proposed by Hu [20], Maitra [21], MacSleyne et al. [22,23,24] and Callahan et al. [25]. Within the field of materials science and simulation of nickel-base superalloys, it has already been successfully applied by Van Sluytman and Pollock to classify various different experimental microstructures [26]. Nguyen et al. also used the method to analyze virtual 3D microstructures from phase-field simulations [27].

In this article, we investigate the shape formation and evolution of tetragonally misfitting precipitate particles upon mutual elastic particle–particle interaction. Experimental studies on γ′′ precipitation in a Nb-containing Ni-based alloy are compared to respective results from phase-field simulations. The discussion is complemented by phase-field simulations on θ′ precipitation in an Al-Cu alloy. Details on the performed experiments are given in Section 2.1. The performance of the phase field simulations is described in Section 2.2. The resulting precipitate shapes are quantitatively evaluated by means of the method on invariant moments, as described in Section 2.3. Section 3 contains the results: first, Section 3.1 explains the experimental studies, followed by Section 3.2, which the phase field simulation studies are presented. In Section 4.1, we discuss the quantitative shape evaluation by means of the method of invariant moments, first the experimental shapes of γ′′ precipitates and then the precipitate shapes obtained from a large-scale 3D phase-field simulation on θ′ precipitation. Finally, the shape evolution of γ′′ precipitates during high-temperature exposure is discussed in Section 4.2.

## 2. Materials and Methods

### 2.1. Experimental Investigations

In order to avoid co-precipitation of the γ′ phase, a special IN718-derivative called alloy 718M was used in this work [28,29], which does not contain γ′-forming elements such as Al or Ti. The alloy has a similar matrix composition compared to the original IN718 alloy, without providing any γ′ precipitation, which is suitable for the investigation of sole γ′′ precipitation [9]. Table 1 shows the average composition of the alloy 718M measured by μ-XRF line scan analysis over the whole cast rods compared to nominal IN718 and 718M.

Single crystalline rods with the composition of alloy 718M (Table 1) were cast in a proprietary Bridgman investment casting furnace (built by chair of Metals and Alloys, University of Bayreuth, Bayreuth, Germany) with a spiral grain selector. The size of the cylindrical single crystalline rod is 16 mm in diameter and 130 mm in length. The temperature gradient and the withdrawal rate of the casting furnace were 6 °C/mm and 3 mm/min, respectively [9,30]. The specimens, adjusted in (001) plane with a thickness of 1 mm, were cut from the single crystal rod with an electrical discharge machine for subsequent microstructure analysis. A solution heat treatment was carried out at 1150 °C for 24 h followed by water quenching. Different isothermal aging treatments at 760 °C and 730 °C for 2 h, 6 h and 10 h were subsequently operated.

Figure 3 shows the range of selected aging parameters, as derived from the isothermal Time Temperature Transformation (TTT) diagram predicted by JmatPro with the databank NiFe-Super (Version 6.1, Sente Software, UK). The aging parameters were chosen to avoid the precipitation of the stable δ phase.

The SEM samples of the alloy 718M were prepared and electrolytic etched in the phosphoric acid solution (3% H_3_PO_4_) at 5 V. Morphology of the precipitates was examined by a SEM 1540EsB (Zeiss, Oberkochen, Germany) with a column-near secondary electron detector (Z-contrast). For a more precise detection of the γ′′ precipitates, thin foils with a diameter of 3 mm for subsequent characterization by transmission election microscopy (TEM) (Zeiss, Oberkochen, Germany) were also prepared from the alloy 718M samples after aging. These TEM foils were electro-polished by the twin-jet method with a solution of perchloric acid and methanol at −20 °C and 20 V. The shapes of the precipitates were determined from dark field images taken with a Zeiss Libra 200 FE transmission electron microscope in <001> zone axis.

### 2.2. Phase-Field Modeling

The phase-field model for the simulation of diffusion-controlled solid-phase precipitation bases on a phenomenological potential functional Ω. It is the volume integral over a microscopic potential density ω(φ), which splits into an interfacial and a bulk contribution:(1)$$\omega_{int}\left(\varphi ,\nabla \varphi \right)+\omega_{bulk}\left(\varphi ,\mu_{i}\right)=\omega_{int}\left(\varphi ,\nabla \varphi \right)+\omega_{el}\left(\varphi ,u_{i}\right)+\omega_{ch}\left(\varphi ,\mu \right),$$
where again the bulk contribution splits into an elastic $\omega_{el}$ and a chemical contribution $\omega_{ch}$. The continuous fields that describe the evolution of the system include the phase-field φ, which discriminates between the fcc matrix (φ=0) and the ordered γ″ phase (φ=1), as well as the chemical potential field μ and the displacement field $u_{i}$.

With respect to interfaces, the sharp phase-field formulation [31,32] is used with a phase-field width of two numerical grid points. The directions used for translational invariance are the <100> directions on a cubic simple finite difference grid. With respect to the boundaries, we apply periodic boundary conditions for all fields. To account for volume change of the simulation domain, we apply non-volume-conserving boundary conditions [5].

#### 2.2.1. Contributions to the Phase-Field Model

The interfacial contribution to the potential functional is taken from Finel et al. [32]. This sharp phase-field model (SPFM) is a naturally discrete formalism that ensures translational invariance of the phase-field profile along a family of directions. We chose the <100> directions of a simple cubic lattice to be translationally invariant. Gradient and Laplacian operators were taken into account up to the third neighbors. The width of the interface was chosen to be twice as large as the grid spacing, and the ponderation coefficients were chosen to be $\gamma_{1}=0.413$ and $\gamma_{2}=0.154$. To interpolate bulk contributions and phase-dependent parameters, we used an interpolation function $h\left(\phi \right)=\phi^{2}\left(3-2\phi \right)$ [31]. The chemical contribution to the grand potential density is given by
(2)$$\omega_{ch}\left(\varphi ,\mu \right)=h\left(\varphi \right)\omega_{ch}^{\gamma^{\prime \prime }}\left(\mu \right)+h\left(1-\varphi \right)\omega_{ch}^{\gamma }\left(\mu \right),$$
where $\omega_{ch}^{\alpha }\left(\mu_{i}\right)$ is the chemical contribution to the potential density of the phase α (γ or γ″). We formulate the multicomponent thermodynamics based on quadratic Gibbs free energy density functions as described in [33]. A concentration-dependent Gibbs free energy density $g_{\mathrm{ch}}^{\alpha }$ is approximated by
(3)$$g_{ch}^{\alpha }=\frac{1}{2}{\left(c_{Nb}-A_{Nb}^{\alpha }\right)}^{2}X^{\alpha }+B^{\alpha },$$
where $c_{\mathrm{Nb}}$ denotes the non-equilibrium *Nb* concentration field and $X^{\alpha }$ the thermodynamic factor matrix, and $A_{\mathrm{Nb}}$ and B are parameters of the parabola [34]. These parameters are temperature dependent and can be calculated from thermodynamic and thermo-kinetic CALPHAD data as described in [33].

The natural variables of the grand potential density are the temperature together with the multiple diffusion potentials μ. They are defined as the partial derivatives of the Gibbs free energy density with respect to the different concentrations. Subsequently, the grand-potential density is obtained from the Gibbs free energy density by means of the following Legendre transformation:(4)$$\omega_{ch}^{\alpha }\left(\mu \right)=g_{ch}^{\alpha }-\mu c_{Nb}\;\mathrm{with}\;c_{Nb}={\left(X^{\alpha }\right)}^{-1}\mu +A^{\alpha }$$

A thorough description of the chemical contribution to this model is given in [33]. The elastic contribution to the grand potential density $\omega_{el}$ is given by
(5)$$\omega_{el}\left(\varphi ,x\right)=\frac{1}{2}\left(\varepsilon -\varepsilon_{0}\right)C\left(\varepsilon -\varepsilon_{0}\right)$$
where ε is the local strain field, $\varepsilon_{0}$ is the anisotropic misfit tensor and *C* is the tensor of elasticity. $\varepsilon_{0}$ and *C* are phase-dependent and therefore implicitly depend on the local coordinate x.

The temporal evolution equations for the state variables are the following:(6)$$\frac{\partial \varphi }{\partial t}=-K\frac{\delta \Omega }{\delta \varphi }\;\frac{\partial }{\partial t}\left(\frac{\delta \Omega }{\delta \mu }\right)=\vec{\nabla }\left(M_{Nb}\vec{\nabla }\mu \right)\;\frac{\delta \Omega }{\delta u_{i}}=\frac{\partial \sigma }{\partial x_{i}}=0$$
where *K* is the phase-field mobility that is chosen to be sufficiently high to ensure purely diffusion-limited kinetics [35], *M_Nb_* is the mobility of *Nb* and σ is the stress tensor. Elastic equilibrium is assumed in every timestep. The set of evolution equations is solved with an explicit Euler scheme.

#### 2.2.2. Phase-Dependent Input Data

The misfit tensor $\varepsilon_{0}$ has three non-zero entries $\varepsilon_{11}{\;=\;\varepsilon }_{22}{\;=\;\varepsilon }_{1}$ and $\varepsilon_{33}{\;=\;\varepsilon }_{3}$. The thermodynamic data is taken from the CALPHAD database TCNi8. Diffusivities of the individual alloying elements are taken from the Dictra database MOBNi4. The interfacial energy is assumed to be isotropic and is estimated from ripening data and from the size dependent aspect ratio of the precipitates respectively [5,35,36,37].

Figure 4a shows the used anisotropic lattice misfits as a function of temperature taken from the literature. Figure 4b shows Density Functional Theory (DFT) data for γ″ from various sources [38,39,40], which are extrapolated by a linear trend. Figure 4c shows elastic constants of single crystalline IN718. The elastic constants of the matrix are obtained by a lever rule as described in [5], where the content of γ″ phase is assumed to be 11%.

The phase-field simulations of θ′ precipitation, discussed here, were performed in conjunction with experimental studies of the mechanical properties of Al alloys by Häusler et al. [42] and Rockenhäuser et al. [43], respectively. As for the simulation of the θ′ phase, first principles data are available for the elastic data of all phases [7,8,44]. Misfit data are available assuming a 3:2 relation between the tetragonal direction of θ′ and the lattice constant of the matrix. The misfit strain is dependent on the stacking of the tetragonal unit cells, which is neglected in this study [45]. The large misfit was found to be in the order of −5 × 10^−3^ [7,46] while the absolute value of the smaller misfit is a factor of 10 smaller. In the case of such a strong tetragonal anisotropy of the misfit, the precipitate aspect ratio only depends on the larger misfit [5].

The circumferential interface of a θ′ precipitate is assumed to be semi-coherent and thus is modeled with a significantly higher interfacial energy. The ratio between coherent and semi-coherent interfaces is reported to be in the range of 2–3. The interfacial energy of the coherent interface is reported to be in the range of 150–250 mJ/m^2^ [7,8,44,47]. Table 2 gives the used material data for the γ″ phase in a Ni-rich matrix at 730 °C as well as for the θ′ phase in an Al-rich matrix at 230 °C.

#### 2.2.3. Simulation Setup

For the 2D simulation of isothermal single variant coarsening of chemo-elastically interacting γ′′ precipitates at 730 °C, the domain size was chosen to be 512 × 512 numerical grid points. This corresponds to a physical domain of 900 nm × 900 nm size. The domain edges are aligned with the crystallographic a- and c-axes of the particles, respectively. The system was initialized with 500 randomly sized precipitates of orientation type [010] with an average aspect ratio 2.6 and a volume between 270 to 360 nm^3^. The area fraction of the γ″ particles was 10%. Periodic boundary conditions were applied at all boundaries and for all state variable fields. The timestep used for the numerical solver was 0.03 s. The initial state of the system was set to be at the physical time *t* = −4 h. Within the first 4 h of simulation duration, the system reaches steady state ripening and overcomes statistical artifacts from the initial state. Thus, at the simulation time 6 h a sufficiently realistic microstructure has evolved from the artificial initial microstructure, which resembles also statistical features from the experimental microstructures. At the simulation time *t* = 6 h, the moment invariants, $\lambda_{1}$ and $\lambda_{2}$, of all the simulated γ″ particles are calculated directly from the phase-field. This is repeated for the simulation time *t* = 10 h to achieve a comparable setting with respect to the experiments. The simulation results are presented in Section 3.2.1.

Next, we discuss the setup for the 3D phase-field simulation of shape formation under elastic particle–particle interaction of the tetragonally misfitting γ″ and θ′ precipitates. To simulate the 3D equilibrium shapes of γ″ and θ′ precipitates, we used a simulation domain of 92 × 92 × 54 numerical grid points. An eighth of a single precipitate particle was introduced with a realistic precipitate volume content of 11.5% for the simulation γ″ precipitates and 2.8% for the simulation of θ′ precipitates, respectively. Mirror symmetries were exploited at the domain boundaries such that only an eighth of the precipitate is modelled [5]. Only interfacial and elastic contributions to the phase-field model were considered, and the system was relaxed until a state of quasi equilibrium was reached. Subsequently, the resulting shapes were evaluated by the method of invariant moments, as will be described in next section, and the results are presented in Section 3.2.2.

### 2.3. Shape-Classification by Invariant Moments

The technologically important mechanism of precipitation strengthening in metallic alloys is also known to be sensitive to the shape of the precipitates. Some shapes turn out to be more effective in hindering dislocation motion through the material than others. Therefore, a systematic and quantitative classification of the shapes of precipitates is useful.

A general way for the quantification of the shapes of precipitates is given by the method of invariant moments [19]. The advantage is that it provides a general framework for the quantitative shape classification and can be applied to arbitrarily shaped particles. Moreover, it is applicable for the characterization of experimental as well as simulated precipitate particle shapes.

As most of the experimental micrographs are two-dimensional (2D) images, here, we restricted it to the 2D case. The calculation of a 2D moment of a precipitate, such as its barycenter, can be easily performed using an indicator or characteristic function, which takes the value one inside the precipitate and zero outside [19]. A 2D moment $\mu_{ij}$ of a precipitate is defined as
(7)$$\mu_{ij}\left(c_{x},c_{y}\right)=\iint_{A}{\left(x-c_{x}\right)}^{i}{\left(y-c_{y}\right)}^{j}\varphi \left(x,y\right)dA,$$
with the spatial coordinates x,y, a reference point $c_{x},c_{y}$ and the characteristic function φ(x,y), which equals 1 inside the precipitate and 0 in the matrix. The order of the moment is given by the sum of the exponents *i* and *j*. The zero order moment $\mu_{00}$ corresponds to the area *A* of the precipitate in 2D. The first-order moments, $\mu_{10}\left(0,0\right),\mu_{01}\left(0,0\right)$, provide the barycenter of the particle. For higher-order moments, we restricted it to the so-called central moments, where the reference point of the moment was chosen to be the barycenter of the particle. Now, we can define two invariants of the second moments:(8)$$\lambda_{1}=\frac{2\mu_{00}^{2}}{\mu_{20}+\mu_{02}},$$
(9)$$\lambda_{2}=\frac{\mu_{00}^{4}}{\mu_{20}\mu_{02}-\mu_{11}^{2}}.$$

The moment invariants *λ*_1_ and *λ*_2_ are invariant during rotation and scaling. From these two invariant moments, it is possible to define a generalized aspect ratio *α_λ_* as follows:(10)$$\alpha_{\lambda }=\frac{\sqrt{\lambda_{2}}+\sqrt{\lambda_{2}-\;\lambda_{1}^{2}}}{\lambda_{1}}.$$

The aspect ratio (*α_λ_*) is calculated as the ratio of the longer axis to the shorter axis, which is therefore *α_λ_* ≥ 1. *α_λ_* = 1 for circle and square. It can be easily shown that for regular shapes, such as a rectangle or ellipse, this generalized aspect ratio *α_λ_* provides the expected value. Moreover, the quantity is well defined for arbitrarily shaped particles.

Further, the quantity *λ*_2_ is invariant under a stretching operation, which for instance would transform a circle into an ellipse. As stretching is a direct operation on the particles’ aspect ratio and λ_2_ remains unchanged upon this operation, this quantity needs to contain other important shape information complementary to the aspect ratio. Thus, it can be regarded as a natural complement to the aspect ratio with regard to the general shape information. However, as *λ*_2_ only takes values in between zero and (4π)^2^ ≈ 157.91 for the shape of ideal ellipses and the value of *λ*_2_ = 144 for an ideal rectangle, we suggest the following normalization:(11)$$Normalized\;\lambda_{2}=\frac{\lambda_{2}^{12}}{{\left(4\pi \right)}^{24}}$$

Then, the normalized λ_2_ has a value of 1 for ellipses/circle and almost 0.33 for rectangular/square shapes. The shape information contained in the aspect ratio is complemented by the normalized λ_2_, which can be understood as general measure of the deviation of the shape of the precipitate from the shape of an ideal ellipse of the same aspect ratio.

Finally, we also provide a consistent way to calculate the major diameter of an arbitrarily shape precipitate particle by means of the method of invariant moments. With α_λ_ provided from Equation (11), the mean major diameter d of an arbitrary 2D shape can be defined by the following equation:(12)$$d=2\times \sqrt{\frac{\mu_{00}\alpha_{\lambda }}{\pi }},$$
where $\mu_{00}$ corresponds to the area of the 2D shape. It is easy to prove that for ideal ellipse and ideal rectangles, this quantity equals to the major diameter and the length of the long edge, respectively.

## 3. Results

### 3.1. Systematic Variation of the Aging Parameters

Figure 5 shows the TEM and SEM microstructures of γ′′ precipitates in the samples of the single crystalline alloy 718M after the different aging treatments, which correspond to parameter setting discussed in Section 2.1. The medians of the plate diameter and the thickness of γ′′ particles are shown in Table 3. Both the diameter and the thickness grow longer with the increased temperature and longer aging time. Due to the lower misfit in the a-axis of the γ′′ phase, the particles grow much longer in planar diameter when the temperature increases or the aging time is longer, whereas the thickness was extended relatively slower. The difference in the growth rate leads to a gradually enlarged median aspect ratio from around 4.6 to 7.8. Note that the thickness of the particles measured by SEM was around 1.2–1.5 times longer than the TEM measurement due to the fact that the interaction volume with electrons in SEM is larger than their actual size. This effect is significant for thickness, and therefore the aspect ratio calculated from SEM measurement is roughly 1.5 times smaller than those from TEM. One can obtain enough data for geometric analysis, especially the in-plane shape, by means of the SEM images from the etched samples, which is relatively difficult in the TEM measurement. However, the length measurement of the precipitates from the TEM images is more precise.

Figure 6a shows the aspect ratio of the γ′′ particles measured from TEM images as a function of the diameter in the samples after aging at 760 °C. The aspect ratio changed from 2 for small particles to around 15 for large particles. With the extension of the aging time from 2 to 10 h, both the major diameter and the aspect ratio of the γ′′ particles became larger. However, the trend gradually leveled over time, which indicated a faster growth in thickness. The phenomenon can be explained by equilibrium shape formation due to the minimization of interfacial and elastic energy in the system [36].

### 3.2. Phase-Field Simulation of Microstructure Evolution during Aging

#### 3.2.1. Development of the Aspect Ratio

The primary parameter to characterize the shape of the tetragonally misfitting γ′′ precipitates is the particles’ aspect ratio. The technical design of aging heat treatments requires a good understanding of the evolution of this shape parameter in γ/γ′′ microstructures upon temperature exposure. We aimed to predict realistic aspect ratios of γ′′ precipitates during the course of technical aging heat treatments using phase-field simulations.

Therefore, the single variant coarsening of chemo-elastically interacting γ′′ precipitates was simulated at 730 °C. In the initial state, at *t* = −4 h, 500 randomly sized particles of orientation type [010] were placed into the system, with an average aspect ratio of 2.6 and an average diameter of 14.3 nm. The microstructures in the initial state after aging for 6 h and 10 h are shown in Figure 7. The coarsening of particles is dominated by a ripening mechanism, where the overall phase fraction remains basically constant and large particles grow at the expense of the smaller ones. However, in later stages, particle coarsening by coalescence events can also be observed. The shapes of the simulated γ″ particles of type [010] were examined by means of the method of invariant moments, as described in Section 2.3.

It is interesting to note that the elastic particle–particle interaction led to a special ordering of the arrangement of the particles. The microstructures after aging for 6 h and 10 h at 730 °C showed a significantly different ordering, compared with the initial state. The particles were predominantly found to be aligned to each other along lines with an angle of about 35° with respect to the (100) direction. This alignment relates to the elastic particle–particle interaction as can be realized by the inspection of the coherence stress distribution. In the bottom row of Figure 7, we show the distribution of the deviatoric elastic stress component, also known as the von Mises stress. The spatial arrangement of precipitates due to elastic interactions has been studied in several previous studies [50,51]. For slow ripening kinetics, the elastic interactions may persist and result in a strong alignment of the precipitates in energetically favorable crystallographic orientation [52,53]. A comparable hexagonal 2D superlattice has been previously identified to be the energetically optimum for γ″ particle arrangement [5].

#### 3.2.2. Study on the in-Plain Shape

The aim of this 3D study was to predict a realistic size dependences of the shapes of elastically interacting precipitates by means of phase-field simulations. The influence of various material parameters including the finite phase fraction on the aspect ratio of γ″ precipitates was studied in [5]. Here, the focus is on the in-plane shape of the precipitates, i.e., the 2D shape that is received upon a cut through a plate-shaped 3D particle in the plane normal to the plate axes (which coincides with the tetragonal c-axes).

In order to efficiently account for realistically strong, elastic particle–particle interactions at finite phase fractions, we proceeded in the same way as described in [5]. We restricted it to a single variant microstructure consisting of a periodic array of particles, as visualized exemplarily in Figure 8b. In case of an arrangement of the particles within a simple tetragonal superlattice, by symmetry it is sufficient to simulate an eighth of a particle. The elastic particle–particle interaction is realized by respective mirror boundary conditions [5].

Figure 8a shows the results of the phase-field simulation study on the in-plane shapes of θ′ precipitates as well as γ″ precipitates. We plot the normalized λ_2_ of the in-plane shape of the precipitates as a function of the particles’ major diameter. For the γ″ particles, we further provide the respective in-plane shapes in black. The in-plane shapes of the θ′ precipitates are omitted, as these are practically circles of different diameters. The precipitate phase fraction was set to 11.5% for the γ″ phase and 2.8% for the θ′ precipitates. Note that the volume fraction of the θ′ phase is substantially lower than that for the other material system. Therefore, the distance between two neighboring θ′ precipitates is substantially larger than the distance between the γ″ precipitates. As a direct consequence of this, the elastic interaction is substantially larger for the γ″ particles. For large γ″ particles, this leads to in-plane shapes that are very different from circular. For particles with a diameter of 100 nm, the in-plane shape even shows small concave fractions. However, despite its complexity, the shape can consistently be classified by the method of invariant moments and specifically by the normalized λ_2_.

## 4. Discussion

### 4.1. Quantitative Shape Classification of Precipitate Particles

#### 4.1.1. Experimental Precipitate Shapes

Figure 9 and Figure 10 show the black/white binarized SEM images from the samples after aging at 730 °C for 6 h and 10 h, respectively. The black regions represent the place where the γ′′ precipitates are located, and the write regions are the γ matrix. The resolution of the images is 2.63 nm/pixel for the sample after 6 h aging (Figure 9) and 5.26 nm/pixel for the sample after 10 h aging (Figure 10). The images were specially taken in an area with separate particles. Therefore, the size of these local particles may be different from the average value. However, the trend of the aspect ratio and the normalized λ_2_ can still be characterized with these local particles. Since the γ′′ phase has three different variants, which indicated three different crystallographic directions of the matrix, the binarized images can also be divided into these three parts: (1) Type [001]: flat-lying particles with a c-axis perpendicular to the image plane; (2) Type [100] and [010]: elliptical particles with the c-axis in the image plane. Some of the unclear particles are eliminated. The moment invariants of the particles in these binarized images were then examined.

Here, the binarization procedure as well as the subsequent classification into the three different orientation types was done by hand, which was a very time consuming task. However, for future studies, it is conceivable to give this task to an appropriately trained convolutional neural network [54].

#### 4.1.2. Simulated Particle Shapes

3D large-scale simulation studies on the θ′ precipitation kinetics have been previously performed [49]. Based on experimentally measured precipitate number densities, a total of 82 precipitates with a random assignment to one of the three possible orientation variants were initially seeded into a 300 × 300 × 300 nm^3^ simulation domain (with a grid spacing of ∆x = 1.5 nm). Periodic boundary conditions were imposed. Then, a phase-field simulation was started accounting for anisotropic interface mobility as well as chemo-mechanical particle–particle interaction with anisotropic and inhomogeneous elasticity. After a simulation time of t = 5 × 10^5^ s, the system reached conditions of steady state ripening. Here, we discuss the phase-field at t = 9 × 10^5^ s.

To be able to quantitatively classify the simulated precipitate shapes, we generated a number a 3 × 100 binarized microstructure images (100 for each spatial direction) from slices from the 3D phase-field simulation data using the visualization software Paraview. In each 2D image, the particles are colored in black and the matrix is indicated in white. The color-jump from white for outside the particles to black for inside the particles is set to be sharp at the phase-field value of φ = 1/2, such that the diffuse interface from the phase-field data is not shown on the microstructure image. Of course, the invariant moments can be calculated also from gray-scale images, showing the original phase-field as gray-scale images of the particles including their diffuse particles/matrix interfaces. However, using the original phase-field values with a diffuse interface, where the color smoothly varies from black to white, provides less accurate values for the invariant moments [19].

The invariant moments, the diameters and the aspect ratio of all the particles on all 3 × 100 binarized microstructure images were calculated, which resulted in a total number of 3003 entries. Focusing on the in-plane shape of the particles, we restricted to those entries with *λ*_1_ > 10, *λ*_2_ > 144 (normalized *λ*_2_ > 0,33) and a diameter larger than 10 nm. Within this data subset, we compared the barycenter of the particles and took from all particles with similar center positions the one entry with the largest surface. This procedure provided 99 entries, which reflects the in-plane shape analysis of the 82 original particles.

In Figure 11, we plot the normalized λ_2_ values from the selected subset of the data as a function of the particles’ major diameter. As expected, for most of the θ′ precipitates, we obtained a normalized λ_2_∼1.0 for their in-plane shape, which corresponds to in-plain shapes very close to a circle. However, some particles also significantly deviate from the circular in-plain shape. It is quite interesting to note that exactly these particles are accidentally subjected to a comparably strong particle–particle interaction, as they all show at least one quite narrow neighbor. Such strong elastic interaction, together with the modified precipitates’ shapes, then influences the kinetics of growth and ripening to a large extent [49].

Two of those groups of three strongly interacting particles are shown in Figure 11. The different 3D particles are indicated by differently colored and partially transparent contour plots. Furthermore, we also show the respective 2D slices from which the in-plain shapes have been derived. The slices are colored white (indicated as light gray) outside the particle and black inside. The dashed line between the particle groups indicates that the plotted distance between the two groups does not correspond to their real distance in the system.

### 4.2. Particle-Shape Evolution during High Temperature Exposure

Upon high temperature exposure, the precipitation microstructure coarsened, i.e., the mean diameter of the particles increased with time. This coarsening behavior was clearly seen in our experiments and is also well captured by the phase-field model (see Figure 5 and Figure 7). Here, we discuss the evolution of the shape parameters, such as the aspect ratio and the normalized λ_2_, as a function of the particle diameter. We compare the results from our experiments, as presented in Section 3.1, with those from the phase-field simulations presented in Section 3.2.

In Figure 12, we compare resulting aspect ratios as a function of the particle’s major diameter from SEM microstructure images, from TEM microstructure images and from phase-field simulations. Furthermore, we provide the size dependence of the aspect ratio [5,35,36], as indicated by the solid black line. There is a very good agreement between the results from the phase-field simulations and the result from the TEM investigations as well as the analytical model by Cozar and Pineau [36]. These results are perfectly consistent with previously published microstructure data on the size dependence of the aspect ratio of the γ″ particles in Nb-containing Ni-based alloys. The results from the SEM micrographs do not agree equally well. As the thickness of the plates is systematically overestimated during the analysis of SEM micrographs, the resulting aspect ratios turn out to be significantly smaller than expected. Nevertheless, the qualitative trend is also correctly captured by our SEM investigations, which are significantly less expensive.

The discussion on the size-dependence of the particles’ aspect ratio is complemented with investigations on the in-plain shape of γ″ precipitates within a wide range of different particles with diameters between 50 nm and 270 nm. In Figure 13, we compare the results from the experimental SEM investigations with those from the phase-field simulation study, as presented in Section 3.2.2. After 6 h of aging, the in-plane shape of the γ′′ particles still have a comparatively round shape with normalized λ_2_ of about 0.6. The value changes after a further 4 h of aging to an average of 0.35, which corresponds to relatively square shape. Note that in this phase-field simulation study, we restricted it to a single variant γ/γ″ microstructure and did not account for the interaction of multiple orientational variants of the γ″ particles. Nevertheless, the agreement is remarkably good. This indicates that the experimentally observed deviations from circular in-plain shapes are predominantly caused by elastic particle–particle interactions.

As described in Section 3.2.1, another possibility to make the aspect ratio bigger is the combination of the γ′′ particles in planar directions. Figure 14 shows the examples when two γ′′ particles contact each other, which results in a sudden increase in both the diameter and the aspect ratio. Such coalescence has not been found in the c-axis, since it is energetically more unfavorable to enlarge in planar directions.

## 5. Conclusions

We investigate precipitation microstructure evolution in metallic alloys by means of experiments and phase-field simulations. Two different, technologically relevant alloys are addressed: γ′′ precipitates in a Nb-containing Ni-based alloy and θ’ precipitates in an Al-Cu alloy. The two material systems both involve tetragonally misfitting precipitates, which exist in three different orientational variants. The focus is on the quantitative classification of the shapes of chemo-mechanically interacting precipitates.
We employed the method of invariant moments, which allowed us to consistently quantify the shapes of arbitrarily shaped precipitates. On this basis, two well-defined quantities were formulated: the generalized aspect ratio and the normalized λ_2_. The latter one complements the shape information given by the first one, as it quantifies the deviation of the shape with respect to an ideal ellipse of a given aspect ratio. These two quantities can be consistently calculated for precipitate particles visible on experimental micrographs as well as particles within phase-field simulation results.We discuss the size-dependence of the precipitates’ aspect ratio for the case of γ′′ precipitates. As shown in Figure 12, we find good agreement between the results from the phase-field simulations, the result from the TEM investigations and an analytical model by Cozar and Pineau [36]. By proper sample preparation techniques, it was further possible to study the shapes of up to 50 nm small γ′′ precipitates on the basis of SEM micrographs. The generation of the latter ones are significantly less expensive than the TEM investigations. It is even possible to quantitatively operate with the shape information gained from SEM micrographs if the systematic underestimation of the aspect ratio of the particles is taken into account.Finally, we study the effect of the elastic particle–particle interaction on the in-plane shape of γ′′ precipitates in Nb-containing Ni-based alloys and θ′ precipitates in Al-Cu-alloys. Here, the in-plane shape denotes the 2D shape received upon a cut through the 3D particle in the plane normal to the tetragonal c-axes. The quantitative reproduction of the experimentally observed in-plane shapes of the γ′′ precipitates requires the elastic particle–particle interaction to be properly taken into account. The interaction scales with the distance and size of the particles and increases naturally for particles with increasing size.

## Figures and Tables

**Figure 1 materials-14-01373-f001:**
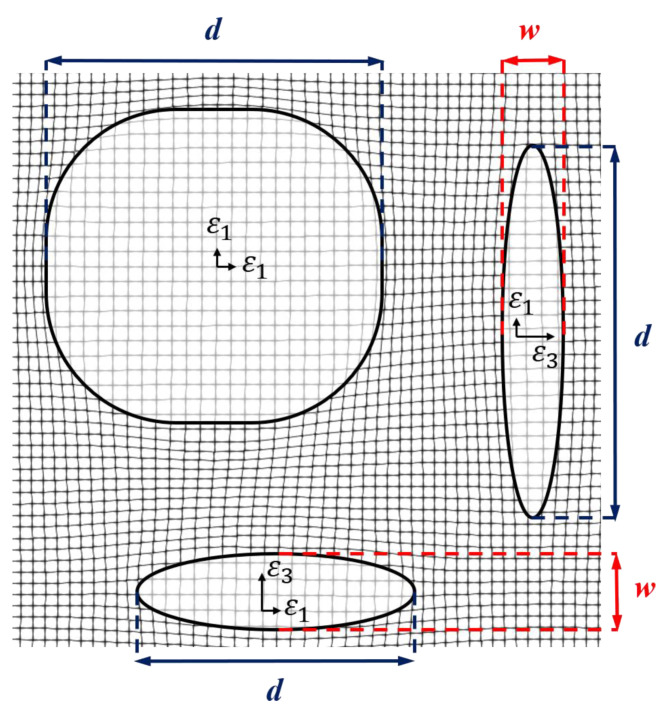
Schematic illustration of three different tetragonally misfitting precipitates embedded in a cubic matrix. The precipitate-matrix interface is indicated by the thick solid black lines. The network of thin black lines indicates the elastic deformation.

**Figure 2 materials-14-01373-f002:**
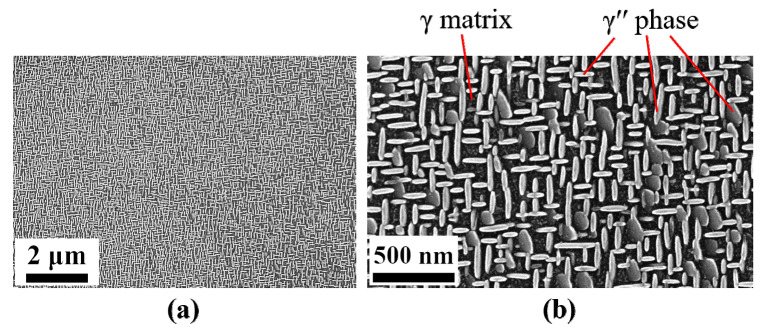
Example SEM images in the interdendritic region from the single crystalline 718M sample after heat treatment. All three orientation variants of the γ′′ particles precipitates contribute to the microstructure.

**Figure 3 materials-14-01373-f003:**
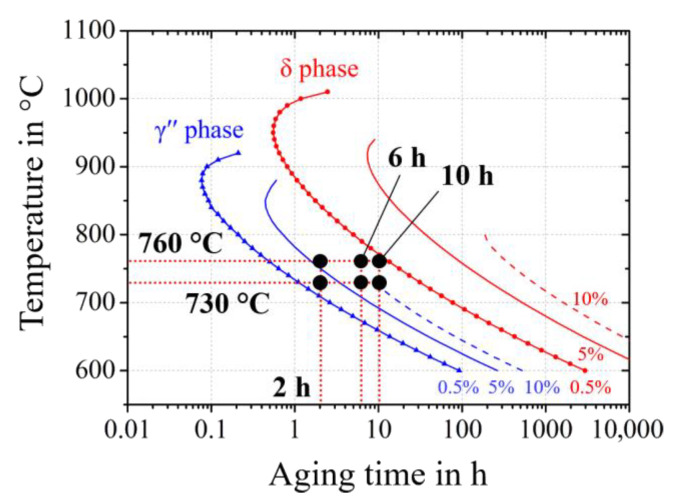
Calculated Time Temperature Transformation (TTT) diagram of the alloy 718M. The curves represent the time passed until 0.5%, 5% and 10% of equilibrium 12 wt.% of γ′′ or stable δ phase. Aging treatments with six different temperature–time combinations are carried out.

**Figure 4 materials-14-01373-f004:**
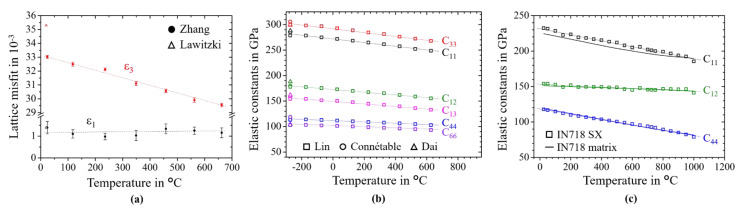
Input parameters used for phase-field simulation as a function of the temperature: (**a**) anisotropic lattice misfit from the literature [6,41]; (**b**) elastic constants of γ″ phase [38,39,40]; (**c**) measured elastic constants of matrix phase.

**Figure 5 materials-14-01373-f005:**
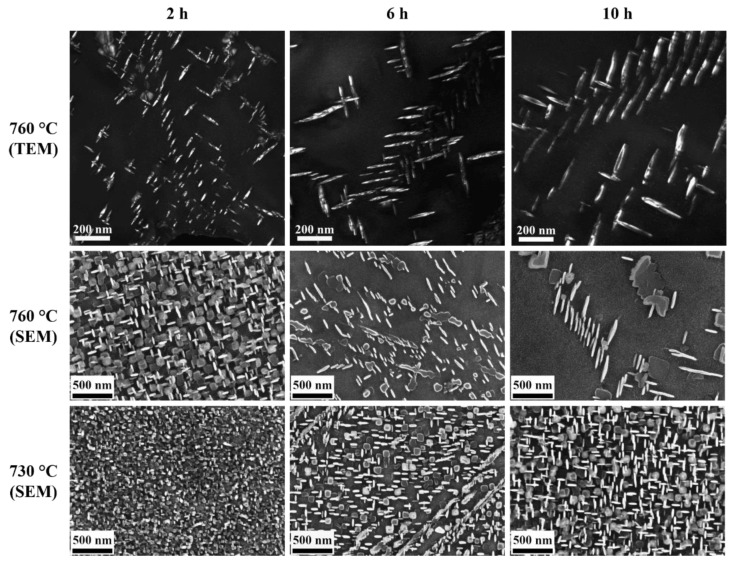
Exemplary TEM and SEM micrographs showing the γ′′ precipitates after aging. The TEM investigations were done on samples aged at 760 °C for 2 h, 6 h and 10 h (top row). SEM studies were performed on samples aged at 760 °C (middle row) and at 730 °C (bottom row) for 2 h, 6 h and 10 h.

**Figure 6 materials-14-01373-f006:**
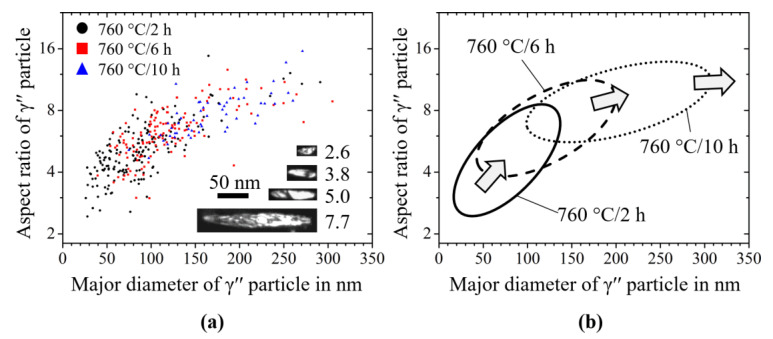
(**a**) Measured aspect ratio of γ′′ particles from the TEM images aged at 760 °C from 2 h to 10 h depending on the major diameter of the particle plate; (**b**) Schematic distribution of these aspect ratios of the γ′′ particles.

**Figure 7 materials-14-01373-f007:**
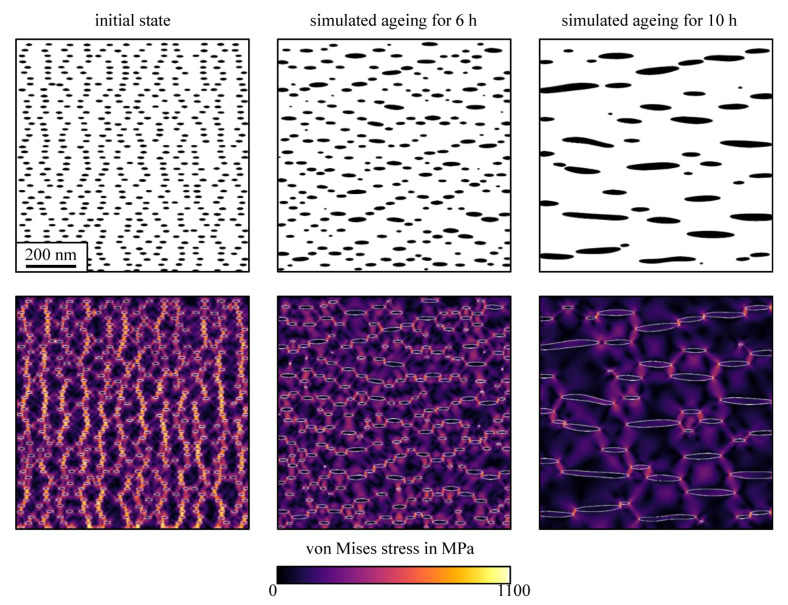
Time series of results from the 2D phase-field simulation of isothermal γ″ particles ripening at 730 °C. In the top row, we show the binarized microstructure, where the particles are indicated in black and the matrix in white. In the bottom row, respective von Mises stress fields are shown. From left to right, we show the initial state, the microstructure after 6 h as well as the microstructure after 10 h.

**Figure 8 materials-14-01373-f008:**
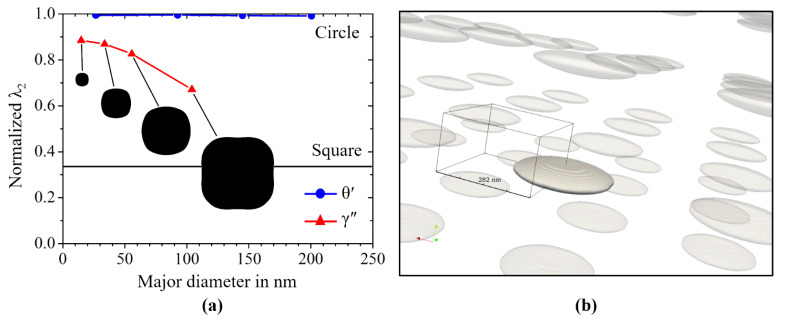
3D simulation study on the in-plane particle shape under elastic particle–particle interaction at constant, realistic phase fractions. (**a**) Plot of the normalized λ_2_ of the resulting in-plain shape of the θ′ precipitates compared to the values for the γ″ precipitates as a function of the major diameter; (**b**) For the visualization of the underlying simulation configuration, the resulting equilibrium shape of θ′ precipitate within the periodically extended-end system of elastically interacting particles is exemplarily shown.

**Figure 9 materials-14-01373-f009:**
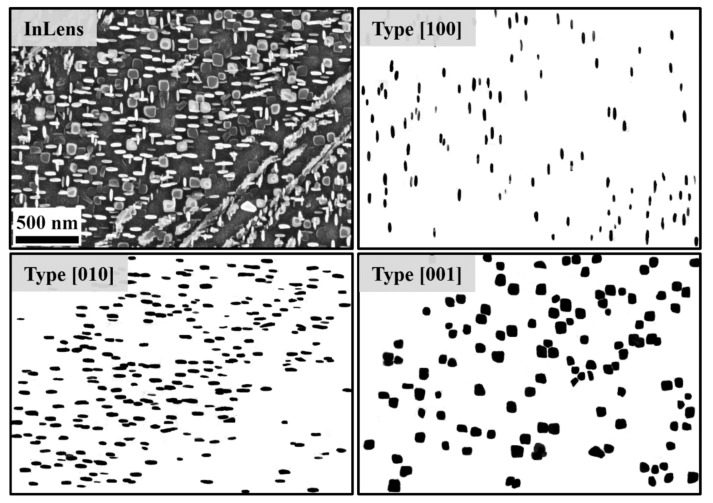
Binarization of the γ′′ particles from a sample after aging at 730 °C for 6 h with regard to three different crystallographic directions.

**Figure 10 materials-14-01373-f010:**
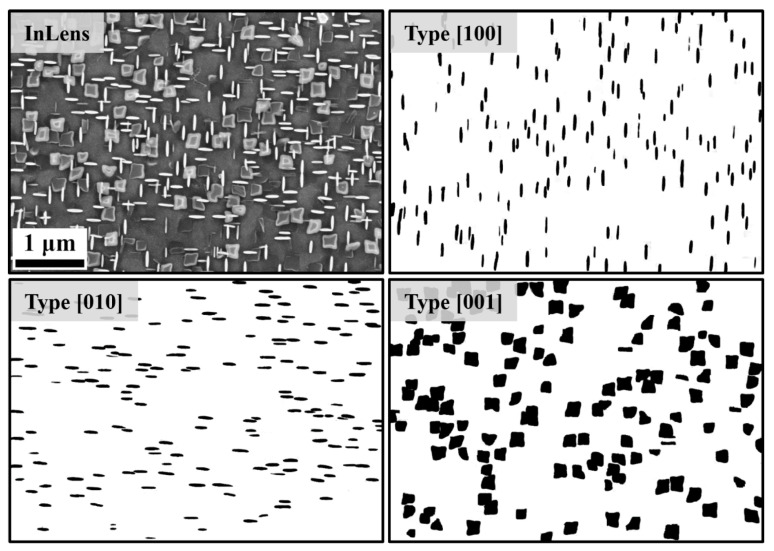
Binarization of the γ′′ particles from a sample after aging at 730 °C for 10 h with regard to three different crystallographic directions.

**Figure 11 materials-14-01373-f011:**
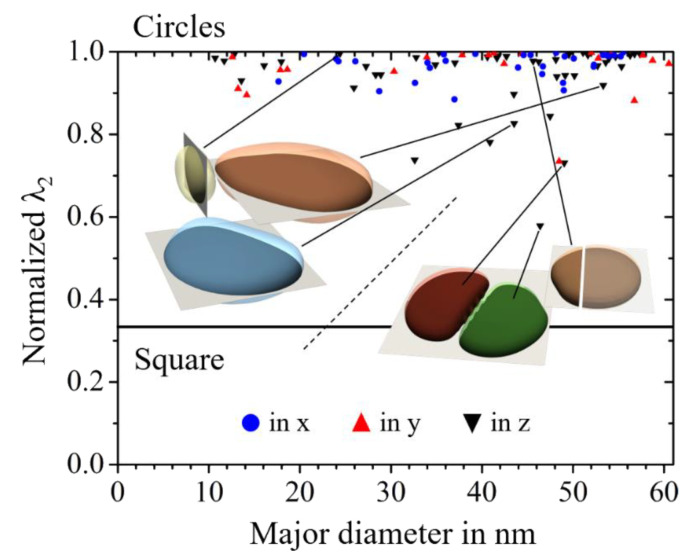
Study on the impact of the particle–particle interaction on the in-plane shape of tetragonally misfitting particles. Plot of the normalized λ_2_ values as a function of the major diameter for the selected data subset from the large-scale 3D simulation study of θ′ precipitation in Al-Cu alloys [49].

**Figure 12 materials-14-01373-f012:**
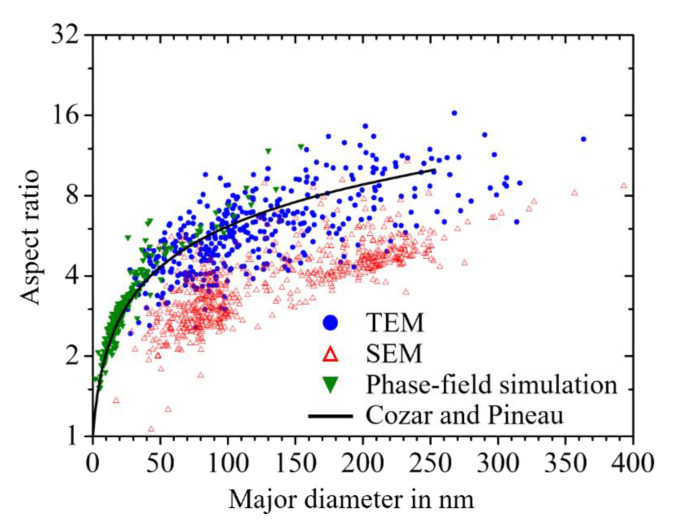
Plot of the aspect ratio from the elliptical site section of γ′′ particle (type [100] and type [010]) as a function of the major diameter. Comparison of SEM results, TEM results, phase-field simulation results and the analytical model of Cozar and Pineau [36].

**Figure 13 materials-14-01373-f013:**
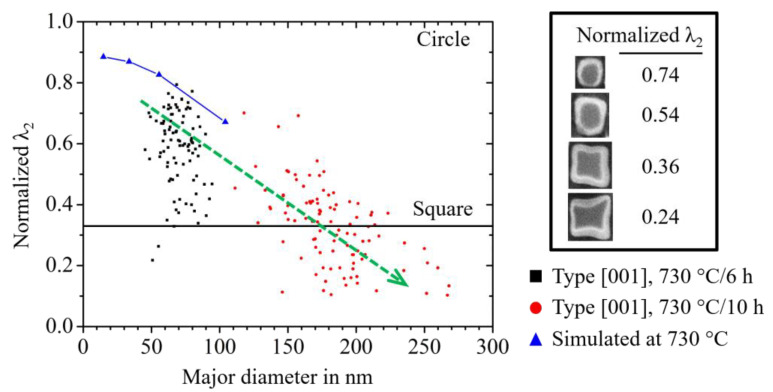
Study on the size-dependence of the in-plain shape of γ″ particles. Plot of the normalized λ_2_ value for particles of orientation type [001] as a function of the major diameter for the SEM-investigations, shown in Figure 9 and Figure 10. This is compared to respective results from the 3D phase-field simulation study presented in Figure 8a.

**Figure 14 materials-14-01373-f014:**
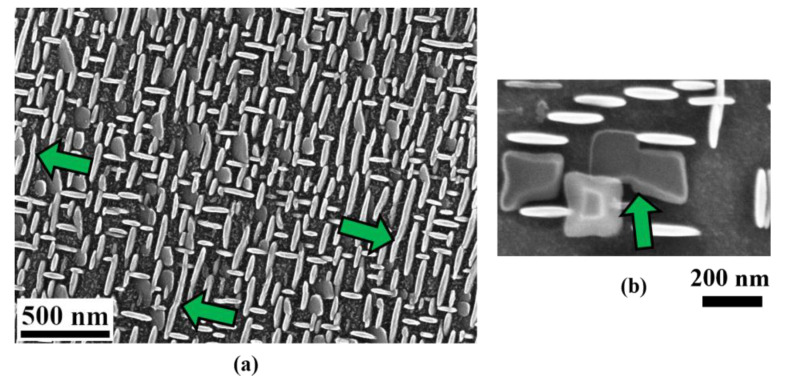
Combination of the γ′′ particles often occurs in planar directions when the volume fraction of the γ′′ phase is very high (see the arrows). (**a**) Normal direction of the γ′′ particles lie in the image plane; (**b**) normal direction of the γ′′ particle is perpendicular to the image plane.

**Table 1 materials-14-01373-t001:** Composition of the single-crystal cast alloy 718M measured by μ-XRF in comparison with commercially available IN718.

Elements in wt.%	Ni	Cr	Fe	Nb	Mo	Al	Ti
IN718	50–55	17–21	Bal.	4.8–5.5	2.8–3.3	0.2–0.8	0.7–1.2
Nominal 718M	58.0	18.0	16.0	5.0	3.0	-	-
Measured 718M	56.3 ± 0.5	18.5 ± 0.4	16.5 ± 0.2	5.5 ± 0.2	3.3 ± 0.1	-	-

**Table 2 materials-14-01373-t002:** Parameters for the phase-field model for γ/γ″ in IN718 at 730 °C and Al/θ′ in an Al/Cu Alloy at 230 °C.

Parameters	γ/γ″ at 730 °C	Al/θ′ at 230 °C	Units
Elastic tensor matrix	C11: 200, C12: 150, C44: 90 [5]	C11: 110, C12: 55, C44: 30 [7]	GPa
Elastic tensor precipitate	C11: 240, C12: 150, C44: 100 C33: 260, C13: 130, C66: 90 [40]	C11: 190, C12: 80, C44: 90 [8]	GPa
Misfit tensor	$\varepsilon_{1}$: $1.22,\;\varepsilon_{3}$: 28.9 [48]	$\varepsilon_{1}$: $7.5,\;\varepsilon_{3}$: −51 [7]	10^−3^
Diffusivity	Niobium: 890; Effective: 180 [ThermoCalc TCNi8]	1.66 [49]	$10^{-20}\;m^{2}\cdot s^{-1}$
Phase-field mobility	10	2.45	$10^{-18}\;m^{2}\cdot s^{-1}$
Interfacial energy density	100 [5,37]	Coherent: 200 Semi-coherent: 400	$mJ\cdot m^{-2}$

**Table 3 materials-14-01373-t003:** Medians of the diameter, thickness and aspect ratio of the γ′′ particles measured by means of TEM and SEM after different aging treatments.

Aging Temperature	Median Diameter in nm			Median Thickness in nm			Aspect Ratio		
	2 h	6 h	10 h	2 h	6 h	10 h	2 h	6 h	10 h
760 °C (TEM)	70	109	174	14	18	24	4.6	6.2	7.8
760 °C (SEM)	65	98	169	22	23	30	3.0	4.2	5.4
730 °C (SEM)	42	81	149	19	23	26	2.2	3.1	3.9

## Data Availability

The research data and parts of the program codes, which are required for the reproduction of the results, can be shared upon reasonable request by the authors. The source codes of phase-field simulation framework will be made publicly available as soon as possible.

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
