# Peer review of "Quantitative Shape-Classification of Misfitting Precipitates during Cubic to Tetragonal Transformations: Phase-Field Simulations and Experiments"

_materials, 2021, doi:10.3390/ma14061373_

Round 1

Reviewer 1 Report

Paper: Quantitative shape-classification of misfitting precipitates during cubic to tetragonal transformations: Phase-field simulations & Experiments present some very interesting results for materials science field.

In my opinion the article is quite large and the authors can consider to present the results (experimental and theoretical) only for NiCrFeNb alloy and γ′′ precipitates – any way the article is focused on γ′′ precipitates analyze.

Few minor corrections:

  • Line 73 and L75: Figure 1 and 2 are the author’s work? than should be placed in section 2 or if are from literature please provide the references.
  • L101: how many determinations were made for chemical composition analyze and what equipment was used (EDS. Spark spectroscopy etc.) for measured 718M.
  • In section 2 please mention information about Al-Cu alloy provenience.
  • L295 6 or 9?

Author Response

Dear Reviewer

Thank you very much for the review of our manuscript. In order to further clarify the focus of our work and to improve the presentation style of the results (as demanded), the introduction has been completely overworked. Furthermore, also the description of the methods (especially the method of invariant moments) and other parts of the manuscript text have been revised.

Please find in the following a list with our responses to your points in red:

  1. In my opinion the article is quite large and the authors can consider to present the results (experimental and theoretical) only for NiCrFeNb alloy and γ′′ precipitates – any way the article is focused on γ′′ precipitates analyze. repl: From our perspective, the presented work focusses on the shapes of tetragonally misfitting precipitates, which applies to both material systems γ’’ precipitates in a Nb-containing Ni-based alloy and θ’ precipitates in an Al-Cu-alloy. This joint focus has been further worked out in the revised manuscript. It is true that, that all the new and original experimental work presented in this manuscript is on γ’’ precipitation. With respect to experimental studies on θ’ precipitation, we refer to previously published work. However, results from original phase-field simulations studies on both different kinds of precipitates are definitely shown and discussed. Our conclusion is based on results from both material systems.
  2. Line 73 and L75: Figure 1 and 2 are the author’s work? than should be placed in section 2 or if are from literature please provide the references. Repl.: The figures 1 and 2 have been created by the authors within the framework of this research work. Both figures are intended to introduce to the topic and to illustrate the manuscripts focus point, i.e. tetragonally misfitting precipitates, their shaping as well as consequences for the resulting precipitation microstructure. Fig. 1 illustrates how tetragonally misfitting precipitates interact with each other via the elastic deformations, that result from the anisotropic misfit. In Figure 2, we exemplarily show the γ/γ’’ microstructure in a Nb-containing Ni-based alloy, which consists of all the three possible orientational variants of γ’’ precipitates at the same time.
  3. L101: how many determinations were made for chemical composition analyze and what equipment was used (EDS. Spark spectroscopy etc.) for measured 718M. Repl.: The composition provided in Table 1 is measured by μ-XRF line scan analysis over the whole single crystal rods, which includes 100 determinations throughout the line. We added some description in the revised manuscript. (line 394-395, page 4)
  4. In section 2 please mention information about Al-Cu alloy provenience. Repl.: We added the references to the publications, which contain further information on the experimental work on θ’ precipitation in section 2.2.2. (lines 513-515, on page 7)
  5. L295 6 or 9? Repl.: This has been clarified in the revised caption of Figure 5.

Reviewer 2 Report

I really liked the article, I haven't read such an interesting article in person for a long time. Congratulations to the authors on a very nice scientific work.

Although I am a reviewer and should be critical of the article, I exceptionally have no comments or suggestions.
Analyzes in determining the morphology of precipitates and changes in lattices are difficult tasks of investigation in the prediction of morphology, formation, distribution ... intermetallics. The use of the phase field method with a vision of the development of intermetallic phases, the method of moment invariants, the method for the analysis of virtual 3D microstructures, the evaluation of the shape parameter in the matrix (in microstructures) as a function of temperature proved to be very suitable. The obtained results are interesting not only for science, but also in application possibilities.

Author Response

Dear Reviewer,

Thank you very much for your kind commendation!

Reviewer 3 Report

There are several issues in the manuscript that should be addressed before further consideration for publication.

  1. Suggest to include discussions on how the solidification rates will affect the precipitates
  2. Suggest to include how these precipitates will affect the material properties such as strength, fatigue.

Author Response

Dear Reviewer:

Gerneral: There are several issues in the manuscript that should be addressed before further consideration for publication.

Repl.: Thank you very much for the review of our manuscript. In order to further clarify the focus of our work and better embed the results into the context of existing research, the introduction has been completely overworked. Furthermore, also the description of the methods (especially the method of invariant moments) and other parts of the manuscript text have been revised.

  1. Suggest to include discussions on how the solidification rates will affect the precipitates Repl.: The influence from solidification rates on the precipitation is certainly another interesting topic. The solidification rates have a strong influence on the distribution of γ″ precipitates, as has been mentioned earlier in DOI: 10.1016/j.matchar.2020.110389. Within the framework of that work, we did not saw any impacts on the shapes of the γ″ precipitates which go beyond the effects of elastic particle-particle interaction, as reported in the current manuscript.
  2. Suggest to include how these precipitates will affect the material properties such as strength, fatigue. Repl.: We agree that it is very interesting to understand the influence of different particles shapes on the resulting mechanical properties. For γ’ precipitates, this has already been investigated by the method of invariant moments in various different alloy systems by Van Sluytman and Pollock in https://doi.org/10.1016/j.actamat.2011.12.008. However, an additional study on the mechanical properties is clearly beyond the scope of the present work.